# A Dataset for Investigating the Impact of Context for Offensive Language Detection in Tweets

**Musa Nuri İhtiyar, Ömer Özdemir, Mustafa Emre Erengül, Arzucan Özgür**

{musa.ihtiyar, omer.ozdemir1, mustafa.erengul, arzucan.ozgur} @boun.edu.tr

Department of Computer Engineering, Boğaziçi University

## Abstract

Offensive language detection is crucial in natural language processing (NLP). We investigated the importance of context for detecting such language in reply tweets on Twitter, where the use of offensive language is widespread. We collected a Turkish tweet dataset where the target group was unvaccinated people during the Covid period. Tweets in the dataset were enriched with contextual information by adding the original tweet to which a particular tweet was posted as a reply. The dataset, which includes over 28,000 tweet-reply pairs, was manually labeled by human annotators and made publicly available. In addition, we compared the performance of different machine learning models with and without contextual information. Our results show that this type of contextual information was not very useful in improving the performance of the models in general, although it slightly increased the macro-averaged F1-score of certain models.

## 1 Introduction

Humans can communicate through language, which enables them to engage in many useful activities, yet language might also be used for destructive purposes. One of the most critical examples of this is offensive language, which can be defined as "any utterance which is blasphemous, obscene, indecent, insulting, hurtful, disgusting, morally repugnant, or which breaches commonly accepted standards of decent and proper speech" (Law-Insider, 2023).

The use of offensive language can occur on a variety of platforms, but is particularly common on online platforms such as Twitter. In recent years, several approaches have been proposed to automatically detect offensive language in tweets. Fine-tuning language models pre-trained with extensive data is considered the current state-of-the-art for detecting offensive language. BERT (Devlin et al., 2019) is one of the most prominent transformer-based pre-trained language models for English and

has also been shown to be very effective in detecting offensive language (Dai et al., 2020; Zampieri et al., 2020; Mozafari et al., 2020). A similar trend can be observed for other languages. For example, Mubarak et al. (2023) used AraBERT (Antoun et al., 2020), the Arabic version of BERT, for Arabic. Similarly, BERTurk (Schweter, 2020) has been successfully used to detect offensive language in Turkish tweets (Beyhan et al., 2022; Toraman et al., 2022; Arın et al., 2023).

Annotated datasets are needed to train or fine-tune machine learning models for offensive language detection. A number of datasets have been prepared for different languages and domains and made publicly available (Basile et al., 2019; Zampieri et al., 2020; ElSherief et al., 2021). A limitation of these datasets is that generally each tweet is labeled individually without considering contextual information. There are few studies that consider contextual information. Mosca et al. (2021) investigate the relative contribution of user information features in machine learning models by using explainability techniques. Cécillon et al. (2021) propose a graph-based approach to represent dialog data from chat logs of an online game and use this representation for abusive language detection. Yu et al. (2022) define context as the previous comment in a Reddit conversation thread and show that such contextual information is useful for detecting hate speech.

We hypothesize that similar contextual information may be useful for offensive language detection in tweets. As a motivating example, consider a reply tweet that states, "I fully agree." The category of this reply tweet (i.e., whether it is offensive or not) depends on the previous context, i.e., the tweet to which it was posted as a reply. To investigate the impact of such contextual information on commonly used machine learning-based offensive language detection models, we collected and manually annotated tweet-reply pairs in Turkish, a

low-resource language with limited datasets. One of the first tweet datasets for detecting offensive language in Turkish was developed by Çöltekin (2020). Recently, Beyhan et al. (2022) and Toraman et al. (2022) also released tweet datasets for Turkish. However, none of these datasets consider contextual information.

We chose our domain as the Covid-19 pandemic, which affected our lives in a number of different ways. Pandemics trigger fear and anger in most people, leading to increased use of offensive language. Sharif et al. (2021) studied the detection of hostile statements in the context of the Covid-19 pandemic, and Bor et al. (2023) showed that such offensive language occurred against unvaccinated people during this period. Therefore, we selected unvaccinated people as our target group.

The main contributions of this paper are twofold: (i) We collect and manually annotate a Turkish tweet dataset specific to the Covid-19 period and containing contextual information in the form of the replied tweet. (ii) We investigate the impact of such contextual information on the performance of commonly used machine learning-based models for offensive language detection. The dataset and source code are made publicly available for future studies.[1]

The rest of the paper is organized as follows. While Section 2 examines the collection and annotation of the dataset, Section 3 focuses on the experiments conducted to compare the machine learning models with and without contextual information. Finally, Section 4 discusses the lessons learned.

## 2  Dataset

We collected a dataset containing replied and reply tweet pairs. A *reply tweet* is a tweet written in response to another tweet, while a *replied tweet* is a tweet to which another tweet has replied. Suppose a tweet $T1$ is posted and then another tweet $T2$ is posted in response to $T1$. In this case, $T1$ is called a replied tweet and $T2$ is called a reply tweet.

Our goal was to create a target group-specific dataset to enable the development of models capable of detecting offensive language towards a specific target group. We selected unvaccinated people in the Covid 19 pandemic as the target group for offensive language. We examined the period from

[1] https://github.com/boun-tabi/CovidOffensiveLanguageUltimateDatasets

March 2020, when the virus reached Türkiye, to September 2022, when the pandemic was no longer on the agenda for most people on the planet. We used search by keyword with 16 different queries such as "aşısız" (unvaccinated) and "aşı olmak istemeyen" (those who do not want to be vaccinated) to identify relevant tweets. The keywords are phrases meaning "aşısız" (unvaccinated) with different singular/plural forms or spellings due to the Turkish character related issues. The list of all keywords used in this study can be found in the Appendix.

There were different options to search for the replied and reply tweet pairs. The first one was getting pairs where at least one of the 16 search keywords occurred in the reply tweet. We call this Dataset 1. Another possibility is that these keywords occur in the replied tweet. This case contains two subcases. The first case is to have at least one of these keywords in a replied tweet, which itself is a reply to another tweet. We refer to this case as Dataset 2. Finally, the last case is to have at least one of these keywords in a replied tweet that is not itself a reply to another tweet. This case is called Dataset 3. All three of these datasets were merged to obtain the final dataset.

Although conversations on Twitter could be arbitrarily long, we only looked at the previous tweet (replied tweet) to avoid unnecessarily complicated data format. In other words, all of the samples in our dataset are a pair. Yet, we could capture any replied-reply couple related to unvaccinated people as long as at least one of the tweets contains one or more of the pre-determined keywords. During the search, we collected tweet ID and tweet text information for both the replied and reply tweets.

Once the collection process was completed, we proceeded with labeling. The objective of the annotation was to obtain a binary label indicating whether or not the reply tweet contains offensive language against unvaccinated people. Making explanations about specific points is essential for this part. First of all, we decided to keep the task clear so that we could understand the impact of the context better, so using a binary label looked like the best option, and we only looked at offensive language against unvaccinated people; in other words, even if a reply tweet was offensive, against immigrants for instance, we labeled that as "not offensive against unvaccinated people" instead of "offensive against unvaccinated people". This was not because such offensive language was acceptable

but due to the fact that we wanted to have a single target group to make the problem more focused such that the effect of the context could be seen more directly. Solely focusing on the offensiveness of the reply tweet was done since the context is relevant only for the reply tweet. That is, a pair where the replied tweet was offensive against unvaccinated people, but the reply tweet was not offensive is categorized as "not offensive" since we are only interested in the reply tweet's behavior.

Which cases to consider as offensive language is another crucial point to explain. Situations like swearing and insulting were the most obvious ones. In addition, provocative words like stating that there should be a punishment, such as not being able to go outside or get into closed areas, without stating any exception or an alternative option, for unvaccinated people are included in this label. Also, we want to express that quotations or simply stating an idea without using harmful language, like saying that "not getting vaccinated is a wrong behavior," are not perceived as offensive language. Even if we determine criteria, as we mentioned, for when to consider a tweet as offensive, this field is inevitably subjective for specific examples. This is why at least two people annotated each pair in our dataset.

The annotation process was carried out as follows. A general guideline for annotation was established and provided to the annotators (i.e., three of the authors of the paper) and a training was performed by using sample examples. Each tweet pair was annotated independently by two annotators and a third annotator was used to resolve inconsistencies. For each tweet pair, there were three label options, namely "not offensive against unvaccinated people", "ambiguous", and "offensive against unvaccinated people". Although it is stated that the goal was obtaining binary labels, three options were given in order to provide more flexibility to the annotators; however, the pairs whose final label is "ambiguous" were removed from the final dataset since this would make the primary goal of the study more difficult to interpret which was examining the effect of taking the replied tweet into account. While doing the annotation, totally unrelated cases in the dataset, such as unvaccinated fruits and vegetables owing to chosen keywords, were mostly cleaned even though a limited number of such cases might be still existing in the dataset. We wanted to measure inter-annotator agreement

for these labels, so we used the F1 and Cohen Kappa scores. We obtained 55.22% and 46.26%, respectively, for these metrics.

After obtaining the annotations by two annotators for each pair of tweets, the annotations were examined. If there is consistency, then this was chosen as the ultimate label. If the ultimate label is "ambiguous", it is removed; otherwise, it is added to the final dataset. If there is inconsistency in the form of one annotator choosing "not offensive" and the other choosing "offensive", these cases are ambiguous; consequently, these were removed as well. For the inconsistencies where one annotator chose "ambiguous", the third annotator looked at the tweet pair and determined the final decision. If a label other than "ambiguous" was chosen, then it is selected as the last label. If not, it was removed. After several hours of this procedure, pairs with binary labels were obtained. In total, we obtained 28808 pairs. While 13478 of them came from Dataset 1, Datasets 2 and 3 contributed with 1515 and 13815 pairs, respectively. The final binary dataset has 27219 examples that are not offensive against unvaccinated people, denoted with 0, and 1589 examples which are offensive against unvaccinated people which are denoted with 2 since 1 was representing the ambiguous case. The dataset is inevitably imbalanced since 94.48% of the pairs are labeled as 0. Inter-annotator agreement for the dataset's last version was measured using the F1 score and Cohen Kappa score. This time they were calculated as 95.21% and 88.97%, which is significantly better than the initial version of the dataset. The final version of the dataset containing the replied and reply tweet ids as well as the manual annotations is made publicly available for future studies.[2]

## 3 Experiments and Results

After completing the annotation of the dataset, we used it to train and evaluate various machine learning models to detect offensive language against unvaccinated people. We randomly selected 20% of the dataset as the test set. For each algorithm we used, we examined two different scenarios. In the first, we used only the reply tweet, while in the second, we studied the impact of using the replied tweet in addition to the reply tweet on our models.

---

[2] https://github.com/boun-tabi/CovidOffensiveLanguageUltimateDatasets

| Method | Prec | Rec | F1 |
|---|---|---|---|
| **KNN (1)** | 20.56 | 41.12 | 27.41 |
| **KNN (2)** | 20.84 | 40.79 | 27.59 |
| **LR (1)** | 50.00 | 39.80 | 44.32 |
| **LR (2)** | 44.72 | 41.78 | 43.20 |
| **MNB (1)** | 65.32 | 26.64 | 37.85 |
| **MNB (2)** | 45.65 | 34.54 | 39.32 |
| **SVM (1)** | 50.76 | 44.08 | 47.18 |
| **SVM (2)** | 51.46 | 34.87 | 41.57 |
| **RF (1)** | 38.51 | 39.14 | 38.82 |
| **RF (2)** | 43.25 | 35.85 | 39.21 |

Table 1: Results for traditional models. For each model, (1) corresponds to the first scenario where only the reply tweet is used and (2) corresponds to the second scenario where both the reply and the replied tweet are used.

| Method | Prec | Rec | F1 |
|---|---|---|---|
| **BERTurk (1)** | 65.73 | 82.68 | 70.28 |
| **BERTurk (2)** | 70.11 | 79.03 | 73.57 |

Table 2: Results for deep learning models. (1) corresponds to the first scenario where only the reply tweet is used and (2) corresponds to the second scenario where both the reply and the replied tweet are used.

## 3.1 Traditional Machine Learning Models

Simple machine learning algorithms might perform quite good for certain tasks. Therefore, we started with simple algorithms such as Logistic Regression (LR), K-Nearest Neighbors (KNN), and Multinomial Naive Bayes (MNB). Then we also used Support Vector Machines (SVM) and Random Forest (RF). Since our dataset was imbalanced, we used downsampling to increase the performance of our models. In other words, we randomly selected a subset for the not offensive class while using all samples for the offensive class since it already had a limited number of samples. We had 1285 positive samples in the training set, so we decreased the not offensive class to 4500 samples, since too much reduction would cause a data scarcity problem. We used a tf-idf based vector representation for the tweets. The performance of the commonly used traditional machine learning algorithms is given in Table 1 with the macro-averaged F1 score, precision, and recall.

There are two main observations we can make with these results. These simple models are not able to perform well on this task. Even if we had used a majority classifier, we would obtain 50.0% recall, 47.24% precision and 48.58% F1 score. The inclusion of information from the replied tweets does not have a significant impact on the performance of the models and behaves more like noise.

## 3.2 Deep Learning Models

Deep Learning models are top-rated in natural language processing. Especially the transformer-based ones (Vaswani et al., 2017) like BERT (De-

vlin et al., 2019) obtained incredible success in the last years. Therefore, we decided to look at the performance of the Turkish version of the BERT model called BERTurk (Schweter, 2020) with and without replied tweet information. For the single tweet setting, we followed the classical procedure for fine-tuning where we used binary cross-entropy with Adam optimizer (Kingma and Ba, 2015) with $5x10^{-5}$ learning rate. We did the hyperparameter optimization by looking at the validation set F1 score. For the case of using two tweets (the reply and replied tweet), the only difference was creating a longer input string by combining the two tweets in the form of "Önceki tweet: replied_tweet Cevap: reply_tweet" (in English, "Previous tweet: replied_tweet Reply: reply_tweet"). The results (macro-averaged scores) obtained on the test set are summarized for the two cases in Table 2.

Interestingly, this time the model that uses both the reply and the replied tweet performed better in terms of F1 score, yet the effect of taking context into account is still limited. Even though precision improves, recall drops. The English translation of an example to explain this phenomenon is provided below. In this example, the reply tweet is offensive, while the replied tweet is not offensive. In this case, including the replied tweet as contextual information to classify the reply tweet misleads the model.

- Replied Tweet: "Vaccination opponents misread the National Anthem."

- Reply Tweet: "Go away the army of brainless people to your village, you can't live in the metropolis without a vaccine."

For more example tweets where the inclusion of context (i.e., the replied tweet) is necessary for the correct classification of the reply tweet and where context could mislead the classifier, see the Appendix.

## 4 Conclusion

We prepared an offensive language dataset for Turkish, where the number of such datasets is very limited. Unlike most other tweet datasets where each tweet is considered individually, we included the replied tweet as contextual information and investigated how this information affects the performance of commonly used machine learning models. Contrary to our expectation, our results showed that this resulted in only a slight improvement in the F1-score for some models and did not significantly improve the performance of the studied models for offensive language detection in general. In theory, the previous tweet appears to contain important information. However, in analyzing our dataset, we found that most reply tweets have only a weak relationship to the replied tweet in terms of meaning. Moreover, interaction with other tweets is dominated by the use of other features on Twitter, such as "like" or "retweet." Consequently, the use of information about previous tweets did not provide much contribution for offensive language detection in this study. Nonetheless, attempting to develop models specifically designed to consider information about previous tweets could lead to better performance and represents a promising future research direction.

## Limitations

While we tried various methods for detecting offensive language with and without replied tweet, we have not focused on developing a specific model which benefits from the previous (i.e., replied) tweet in the best way. Our goal was to investigate the impact of contextual information on the performance of commonly used machine learning-based models. Therefore, even though we were not able to get significant improvements with contextual information, further research focusing on this subject is a promising direction to follow.

We examined the use of previous tweet for only single target group and language due to the laborious nature of the manual annotation process and the time limitations. The dataset can be expanded with other target groups and languages in the future.

## Ethics Statement

Offensive language detection systems could be very useful for real-life uses. Because machine learning-based models are guided mainly by the data they use, the annotation of datasets is an essential step, which ought to be carried out responsibly. Despite the fact that we tried to use multiple annotators for the labeling process, developing better strategies are possible since some examples regarding offensive language are very subjective. The annotated data is shared based on Twitter's terms of use.

## Acknowledgements

This work is partially supported by the EU funded project entitled "Utilizing Digital Technology for Social Cohesion, Positive Messaging and Peace by Boosting Collaboration, Exchange and Solidarity" and by the Boğaziçi University Research Fund under the Grant Number 16903.

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

# A    Tweet Pair Examples Regarding Context Information

## A.1    An example where context is necessary for correct classification of the reply tweet

The English translation:

Replied Tweet: If we are closed at home again because of those who are not vaccinated, you will see curses that you have not seen so far in this account..

Reply Tweet: +1

## A.2    An example where context does not matter

English translation:

Replied Tweet: It may be against the necessity of vaccination, it may be thought that the mask is not protective; however, there is no human side of walking as a group on a girl who works as a cashier under difficult conditions, entering a closed area without a mask, and causing fear and sadness.

Reply Tweet: Those who are not vaccinated + those who do not wear masks. I seriously don't understand what's wrong with this team. This team is seriously litmus of intelligence.

## A.3    An example where reply is not offensive but replied might mislead since it is offensive

English translation:

Replied Tweet: Prof. Bingür Sönmez: Those who say they will not get vaccinated are traitors, we will not allow them to get married with our girls

Reply Tweet: At the point where the cardiovascular surgeon has come, we will not allow traitors who do not get vaccinated to get married with our girls.

### A.4 An example where reply is offensive but replied might mislead since it is not offensive

English translation:

Replied Tweet: Vaccination opponents misread the National Anthem

Reply Tweet: Go away the army of brainless people to your village, you can't live in the metropolis without a vaccine.

## B Keywords used for Getting Related Tweets

The following keywords were used in our search: aşısız, asısız, aşısızlar, asısızlar, aşı olmayan, ası olmayan, aşı olmayanlar, ası olmayanlar, aşı olmak istemeyen, ası olmak istemeyen, aşı olmak istemeyenler, ası olmak istemeyenler, aşı yaptırmayan, ası yaptırmayan, aşı yaptırmayanlar, ası yaptırmayanlar.

