# OpenReview forum: "A Dataset for Investigating the Impact of Context for Offensive Language Detection in Tweets"
_EMNLP/2023/Conference — EMNLP 2023 Findings_

### Official Review · Reviewer_UEvi · 2023-08-05

**Soundness:** 4

**Excitement:**

4: Strong: This paper deepens the understanding of some phenomenon or lowers the barriers to an existing research direction.

**Paper Topic And Main Contributions:**

In this work, the authors developed a Turkish dataset for for investigating the impact of context for offensive language
detection in tweets. The dataset creation is well documented and is a valuable resource as there aren't many datasets for this domain in the Turkish language. The dataset task is well evaluated with traditional methodologies.

**Reasons To Accept:**

- Dataset in a low-resource language
- Decent manual curation results

**Reasons To Reject:**

- Positive class is less than 5% of the total dataset. Making the usability of the resource limited.

**Reproducibility:**

2: Would be hard pressed to reproduce the results. The contribution depends on data that are simply not available outside the author's institution or consortium; not enough details are provided.

**Reviewer Confidence:**

4: Quite sure. I tried to check the important points carefully. It's unlikely, though conceivable, that I missed something that should affect my ratings.

---

> ### Author Rebuttal · Authors · 2023-08-29
>
> We thank the reviewer for the valuable comments. We briefly address the comments and questions below.
>
> **Q:** “Positive class is less than 5% of the total dataset. Making the usability of the resource limited.”
>
> **A:** This is correct, yet we collected as much data as we could and the nature of the dataset turned out to have this skewness. Nevertheless, there are over 1500 tweets in the positive class and we believe the dataset would be a useful resource for further studies, not only for NLP, but also for social and health sciences.

---

### Official Review · Reviewer_tdPF · 2023-08-05

**Soundness:** 3

**Ethical Concerns:**

Yes

**Excitement:**

3: Ambivalent: It has merits (e.g., it reports state-of-the-art results, the idea is nice), but there are key weaknesses (e.g., it describes incremental work), and it can significantly benefit from another round of revision. However, I won't object to accepting it if my co-reviewers champion it.

**Justification For Ethical Concerns:**

The release of such data where individual tweets are labelled by a negative label (i.e., offensive language) can be ethically questionable if the identity of the tweet author is known. The paper stated that the tweet IDs will be released which means tweet author identity can be easily found. Authors didn't comment on this.

**Paper Topic And Main Contributions:**

The paper describes the construction of a dataset of 20K Turkish tweet-reply pairs manually annotated by offensive language presence. The problem of offensive language detection is also tackled by comparing the performance of different machine learning models with and without contextual information.

**Reasons To Accept:**

-	The developed dataset offers annotated tweets with contextual information which is different from existing Turkish datasets. The data is also large in scale.

**Reasons To Reject:**

-	Some decisions made for dataset construction don’t seem reasonable/justified to me. For example, the authors limited the definition to offensive/non-offensive labels to a certain aspect of a topic (against covid vaccinated people) (lines 159 – 168) and even if a tweet is offensive but not fitting this criterion, it was labelled as non-offensive. I think this limited the scope and applicability of the dataset greatly, especially with the COVID-19 topic becoming almost obsolete these days. Such dataset can also be very confusing to machine learning models, where the same language (e.g., offensive language is usually associated with common phrases) can appear in both offensive and non-offensive tweets (Since offensiveness is topic-dependent according to the data annotation guidelines).
-	Due to the extreme skewness of the dataset (only 5% of the dataset has a positive label), this means the way the test set was constructed (randomly sample), it will be easy to defeat by a majority label baseline. Such model should have been reported in the paper. Moreover, this raises the question on how much we can learn about the performance of the provided models on such test set. Such data skewness can also be relevant to the first weakness I reported.

**Reproducibility:**

4: Could mostly reproduce the results, but there may be some variation because of sample variance or minor variations in their interpretation of the protocol or method.

**Reviewer Confidence:**

4: Quite sure. I tried to check the important points carefully. It's unlikely, though conceivable, that I missed something that should affect my ratings.

**Typos Grammar Style And Presentation Improvements:**

-	Sec 2 could have really benefited from real tweet examples from the dataset.
-	In Sec 2, there are several types of tweets (reply tweets, replied tweets), please define each clearly.
-	The difference between Dataset 3 and the other two isn’t clear. How can a reply tweet not reply to any other tweet? “replied tweet contains these keywords, yet it is not a response to any other tweet”
-	Lines 240-243: Since in case of disagreement between the first two annotators, these pairs were dropped (Except for ambiguous), then it is normal to have such high agreement compared to stage 1. Also, providing stats on labels for every annotation stage would have been useful.

---

> ### Author Rebuttal · Authors · 2023-08-29
>
> We thank the reviewer for the valuable comments and feedback to further improve the paper. We briefly address the comments and questions below.
>
> **Q:** “Some decisions made for dataset construction don’t seem reasonable/justified to me. For example, the authors limited the definition to offensive/non-offensive labels to a certain aspect of a topic (against covid vaccinated people) (lines 159 – 168) and even if a tweet is offensive but not fitting this criterion, it was labelled as non-offensive. I think this limited the scope and ap-plicability of the dataset greatly, especially with the COVID-19 topic becoming almost obsolete these days.”
>
> **A:** Our aim was to create a target-specific data set, where the target group for offensive language is unvaccinated people in the Covid-19 pandemic. So, rather than detecting offensive language, our goal was to enable the development of models that are able to detect offensive language towards a certain target, which is a more challenging task than detecting the existence of offensive language in general. We will explain this more clearly in the paper.
> We chose the Covid-19 topic, because we believe infectious diseases and vaccination/unvaccination will continue to be important topics and the created dataset may be useful for further analysis in other disciplines as well, such as social or health sciences.
>
> **Q:** “Due to the extreme skewness of the dataset (only 5% of the dataset has a positive label), this means the way the test set was constructed (randomly sample), it will be easy to defeat by a majority label baseline.”
>
> **A:** We thank the reviewer for this comment. We will report the performance of the majority baseline in the revised paper. Such a baseline will not be able to identify the positive class (minority class). We will also report the performance separately for each class (positive and negative) to demonstrate the success of the classifiers for both the majority and the minority classes.
>
> **Q:** “Sec 2 could have really benefited from real tweet examples from the dataset.”
>
> **A:** We have sample tweet pairs in the Appendix, but we will also include examples in Section 2 to en-hance the definitions and descriptions. We thank the reviewer for this suggestion.
>
> **Q:** "Dataset 3 and the other two isn’t clear. How can a reply tweet not reply to any other tweet?"
>
> **A:** It is not “a reply tweet” not reply to any other tweet, but instead pairs where “the replied tweet” contains these keywords, yet it is not a response to any other tweet. We agree that this explanation is confusing, and we will clarify this with an example in the revised paper. In Dataset 2, the “replied” (that is original) tweet contains these keywords, and that tweet itself is a reply to another tweet. In Dataset 3, that tweet is not a reply to another tweet. Suppose we have three tweets T1, T2 and T3 and suppose, T2 is a reply to T1 and T3 is a reply to T2 (i.e.,  T1 > T2 > T3). In that case, T2 is included to Dataset 2, whereas T1 to Dataset 3 if they contain at least one of the keywords. Note that T1 is not a response to any other tweet, it is just a replied tweet. On the other hand, T2 is a reply tweet to T1, but replied tweet for T3.
>
> **Q:** "there are several types of tweets (reply tweets, replied tweets), please define each clearly."
>
> **A:** We thank the reviewer for pointing this issue, we will define each in the revised paper as follows:
> - reply tweet: A tweet which is written as an answer to another tweet
> - replied tweet: A tweet which was answered by another tweet
>
> **Q:** "Since in case of disagreement between the first two annotators, these pairs were dropped (Except for ambiguous), then it is normal to have such high agreement compared to stage 1."
>
> **A:** Yes, in fact, our goal was obtaining a dataset with a reliable level of consistency; therefore, we fol-lowed such a strategy.
>
> **Q:** "Also, providing stats on labels for every annotation stage would have been useful."
>
> **A:** We thank the reviewer for this comment. We will include these statistics to the revised paper.
>
> **Justification For Ethical Concerns:** We thank the reviewer for pointing this issue and we will dis-cuss this in the Ethics Statement section of the paper. Unfortunately, we do not know any alternative way to develop a public dataset related to offensive language detection in tweets. Due to the terms of use of Twitter we are not allowed to publish the tweets themselves, but only the tweet IDs.

---

### Official Review · Reviewer_xKL1 · 2023-08-11

**Soundness:** 2

**Excitement:**

2: Mediocre: This paper makes marginal contributions (vs non-contemporaneous work), so I would rather not see it in the conference.

**Paper Topic And Main Contributions:**

The paper presents a Turkish dataset of pairs of tweets where one tweet is a reply to the other one. The dataset was manually annotated with binary labels (for "offensive language against unvaccinated people" or not) by the authors. The paper also reports the performance of classifiers based on the basic machine learning models and the collected dataset. This papers reports negative results.

**Questions For The Authors:**

Could you add full list of keywords? (since it's only 16 of them)

**Reasons To Accept:**

1. Turkish is a low resource language. Hence, having more human annotated datasets can potentially be valuable.
2. Reporting negative results can potentially be helpful for other scientists.

**Reasons To Reject:**

The paper has significant amount of scientific errors as well as missing to convey the significance of the results.

1. Without full list of keywords (which contains just 16 of them, so could be easily provided), it is difficult to understand what kind of data is being extracted in Tweets.
2. It seems like the authors were the annotators of the dataset. This could be fine for training set, but inappropriate for test set. The papers states that the authors randomly selected 20% of the self-annotated dataset. This is problematic. Tests set should be a separate, never seen, independently annotated set.
3. It appears that the third author was used as an adjudicator for situation where the other two annotators disagreed. It would be more appropriate ti have an independent, trained adjudicator.
4. Results are hard to evaluate (and to replicate). No details about the fine-tuning of the models were provided. Also, it is unclear how the data was fed to the transformer (this may change how we can interpret the results)
5. Overall scientific value of the paper is low (since the annotation and the adjudication was done by the authors).

**Reproducibility:**

3: Could reproduce the results with some difficulty. The settings of parameters are underspecified or subjectively determined; the training/evaluation data are not widely available.

**Reviewer Confidence:**

4: Quite sure. I tried to check the important points carefully. It's unlikely, though conceivable, that I missed something that should affect my ratings.

**Typos Grammar Style And Presentation Improvements:**

The language of the paper is rather informal. Some parts of the paper are confusing for understanding (dataset section).

---

> ### Author Rebuttal · Authors · 2023-08-29
>
> We thank the reviewer for the valuable feedback. We briefly address the comments and questions be-low.
>
> **Q:** “Without full list of keywords (which contains just 16 of them, so could be easily pro-vided), it is difficult to understand what kind of data is being extracted in Tweets.”
>
> **A:** The keywords are phrases having the meaning “aşısız” (unvaccinated) with singular-plural form or spellings due to the Turkish character related issues. Here is the full list of keywords, these will also be included to the Appendix of the paper. We thank the reviewer for pointing this.
>
> aşısız
>
> asısız
>
> aşısızlar
>
> asısızlar
>
> aşı olmayan
>
> ası olmayan
>
> aşı olmayanlar
>
> ası olmayanlar
>
> aşı olmak istemeyen
>
> ası olmak istemeyen
>
> aşı olmak istemeyenler
>
> ası olmak istemeyenler
>
> aşı yaptırmayan
>
> ası yaptırmayan
>
> aşı yaptırmayanlar
>
> ası yaptırmayanlar
>
>
> **Q:** “It seems like the authors were the annotators of the dataset. This could be fine for train-ing set, but inappropriate for test set.”
>
> **A:** The main contribution of the paper is the annotated data set and the annotation process was time consuming and laborious. Therefore, the annotators were included as authors to the paper due to their significant annotation efforts. A general guideline for annotation was determined and provided to the annotators and a training was performed by using sample examples. As explained in the Dataset sec-tion, each tweet was annotated by two annotators independently, a third annotator was used to resolve inconsistencies and tweet pairs with ambiguity were filtered by following the procedure described in the Dataset section of the paper. The inter-annotator agreement F1 score and Cohen Kappa score for the final version of the dataset was calculated as 95.21% and 88.97%, respectively.
> The models used in the paper are standard models with the main goal of providing a baseline for in-vestigating the impact of context. Feature engineering was not performed; thus, there was no any an-notator input injected into the models.
>
> **Q:** “Results are hard to evaluate (and to replicate). No details about the fine-tuning of the models were provided. Also, it is unclear how the data was fed to the transformer (this may change how we can interpret the results)”
>
> **A:** We thank the reviewer for this comment. We will include these details to the final version of the paper.

---

### Official Review · Reviewer_xHbP · 2023-08-13

**Typos Grammar Style And Presentation Improvements:** None
**Soundness:** 3

**Excitement:**

3: Ambivalent: It has merits (e.g., it reports state-of-the-art results, the idea is nice), but there are key weaknesses (e.g., it describes incremental work), and it can significantly benefit from another round of revision. However, I won't object to accepting it if my co-reviewers champion it.

**Missing References:**

None

**Paper Topic And Main Contributions:**

The paper discuss offensive language detection in reply of tweets with a focus on the context importance.

The main contributions of the paper are:

1. The creation and public release of a new dataset: The authors collected a Turkish tweet dataset targeting unvaccinated people during the Covid period. This dataset consists of over 20,000 tweet-reply pairs, where the replies are enriched with contextual information by adding the original tweet to which a particular tweet was posted as a reply. The dataset was manually labeled by human annotators.

2. An empirical study comparing the performance of different machine learning models for offensive language detection with and without the provided contextual information. The results showed that while the contextual information slightly increased the macro-averaged F1-score of certain models, it generally did not significantly improve the performance of the models.

Areas of improvement:
-  Given the limited impact of context in this study, it might be interesting to explore other forms of context, such as user profiles, user history, or even the broader conversation thread.
- Bias Analysis: A deeper dive into potential biases introduced by the choice of target group and how they might be mitigated in future studies would be beneficial.


**Questions For The Authors:**

- How were the human annotators trained to label the tweets, and how did you ensure consistency in their annotations?
- Could you elaborate on why the contextual information did not significantly improve model performance in most cases?
- Were there specific types of offensive language or subtleties that the context helped to identify?
- Why was the specific target group of unvaccinated people during the Covid period chosen for this study?

**Reasons To Accept:**

The major contribution is three fold. 1) Dataset Introduction 2) Context-wise Enrichment 3) Comparative Analysis

Dataset Introduction: The data would be a valuable contribution to the community. The authors collected 20,000 tweet-reply pairs and manually label them. This dataset can serve a test bed for Turkish NLP tasks.

Context-wise Enrichment: The authors enhanced the original tweets with corresponding replies. This thread based dataset can potentially help in understanding the nuances and subtleties of offensive language in a conversation.

Comparative Analysis: The comparison of machine learning model performances with and without contextual information provides a clear benchmark for future studies.


**Reasons To Reject:**

Contextual Impact: The paper concludes that the contextual information was not very useful in improving the performance of the models. It would be beneficial to delve deeper into why this was the case. Is it because of the nature of the tweets, the models used, or some other factor?
Model Details: The summary does not provide details about which machine learning models were used. A more detailed analysis of the models, their architectures, and hyperparameters would be beneficial for reproducibility and a deeper understanding of the results.
Target Group: While the choice of focusing on unvaccinated people during the Covid period is unique, it might also introduce biases. The paper should discuss potential biases and how they might affect the results.
Macro-averaged F1-score: The paper mentions a slight increase in the macro-averaged F1-score for certain models with context. It would be helpful to know which models benefited from the context and why.


**Reproducibility:**

5: Could easily reproduce the results.

**Reviewer Confidence:**

4: Quite sure. I tried to check the important points carefully. It's unlikely, though conceivable, that I missed something that should affect my ratings.

---

> ### Author Rebuttal · Authors · 2023-08-29
>
> We thank the reviewer for the valuable comments and constructive feedback. We briefly address the comments and questions below.
>
> **Q:** “Macro-averaged F1-score: The paper mentions a slight increase in the macro-averaged F1-score for certain models with context. It would be helpful to know which models benefited from the context and why.”
>
> **A:** The results in Tables 1 and 2 show that MNB, RF, and BERTurk benefited from the context information. However, the traditional ML models including MNB and RF obtain much lower performance compared to BERTurk with or without context information. We investigated the results obtained by BERTurk in more detail and identified three main types of cases where context information is helpful or where it misleads the model. These three cases are listed with one sample example each in the Appendix section.
>
> **Q:** How were the human annotators trained to label the tweets, and how did you ensure consistency in their annotations?
>
> **A:** A general guideline for annotation was determined and provided to the annotators and a training was performed by using sample examples. As explained in the Dataset section, each tweet was anno-tated by two annotators independently, a third annotator was used to resolve inconsistencies and tweet pairs with ambiguity were filtered by following the procedure described in the Dataset section of the paper. The inter-annotator agreement F1 score and Cohen Kappa score for the final version of the da-taset was calculated as 95.21% and 88.97%, respectively.
>
> **Q:** Could you elaborate on why the contextual information did not significantly improve model performance in most cases?
>
> **A:** We were expecting the previous (original or replied) tweet to contain important information related to the class of the reply tweet. However, by analysing the results and the dataset, we found that the reply tweet in many cases is not a direct response to the original tweet, but rather provides the author’s independent opinion on the topic. Also, in some cases, while the reply tweet is in one category, the original tweet may be of the opposing category misleading the models. Example cases are provided in the Appendix. As the reviewer suggested, including other types of context information may be more helpful, which we plan to investigate as future work. Nevertheless, we believe the current dataset and the results will provide a useful basis for future work in this direction.
>
> **Q:** Were there specific types of offensive language or subtleties that the context helped to identify?
>
> **A:** We provided example cases in the Appendix, but we will conduct a more detailed analysis. We thank the reviewer for this point.
>
> **Q:** Why was the specific target group of unvaccinated people during the Covid period chosen for this study?
>
> **A:** We believe infectious diseases and vaccination/unvaccination will continue to be important topics and the created dataset may be useful for further analysis in other disciplines as well, such as social or health sciences.

---

### Meta-Review · Area_Chair_oZ3y · 2023-09-22

**Recommendation:** 2

**Metareview:**

This study proposes a dataset, which comprises of 20K+ tweet-reply pairs. To enhance contextual understanding, each reply incorporates the original tweet to which it was posted in response. Human annotators carefully labeled the dataset. This study also provides experimental results with and without contextual information.

Among reviewers there is a consensus about the dataset as it provided replies as context. However, there are concerns about the annotation process, utilization of context and so on.

Reviewer xHbP raised concern about the benefit of contextual information. Reviewer tdPF questioned about the skewed distribution issue in the dataset. While authors addressed the concern, however, reviewer still has concerns such as data distribution and data sharing, which reflected in the rebuttal.  Reviewer UEvi also raised the same concern regarding the data distribution.

---

### Decision · Program_Chairs · 2023-10-07

**Decision:**

Accept-Findings

**Comment:**

This study proposes a dataset, which comprises of 20K+ tweet-reply pairs. To enhance contextual understanding, each reply incorporates the original tweet to which it was posted in response. Human annotators carefully labeled the dataset. This study also provides experimental results with and without contextual information.

Among reviewers there is a consensus about the dataset as it provided replies as context. However, there are concerns about the annotation process, utilization of context and so on.

Reviewer xHbP raised concern about the benefit of contextual information. Reviewer tdPF questioned about the skewed distribution issue in the dataset. While authors addressed the concern, however, reviewer still has concerns such as data distribution and data sharing, which reflected in the rebuttal.  Reviewer UEvi also raised the same concern regarding the data distribution.